# Could Microplastics Be a Driver for Early Onset Colorectal Cancer?

**DOI:** 10.3390/cancers15133323

**Published:** 2023-06-24

**Authors:** Shelley Li, Jacqueline I. Keenan, Ian C. Shaw, Frank A. Frizelle

**Affiliations:** 1Department of Surgery, University of Otago Christchurch, Christchurch 8011, New Zealand; lish6670@student.otago.ac.nz (S.L.); jacqui.keenan@otago.ac.nz (J.I.K.); 2School of Physical & Chemical Sciences, University of Canterbury, Christchurch 8041, New Zealand; ian.shaw@canterbury.ac.nz

**Keywords:** colorectal cancer, microplastics, early onset colorectal cancer, colonic mucus layer

## Abstract

**Simple Summary:**

The incidence of colorectal cancer has increasing dramatically in those under 50 years of age. These cancers are prominently in the sigmoid colon and rectum. This increase in incidence of colorectal appears his appears to have be a cohort effect impacting on those born after 1960, and to be compounding with time. The speed and size of this increase in incidence means that this is thought to be a result some change in an environmental factor as opposed to an inherited genetic change. We explore a possible role of microplastics as a driver for this phenomena. We discuss the possible role that microplastics may play, and mechanism for such. We discuss how their composition, and/or effect on barriers function of colonic mucus and their impact on the colonic mucosa, and result in increased exposure to the faecal matter and their associated bacteria and toxins.

**Abstract:**

**Introduction:** The incidence of colorectal cancer in those under 50 years of age (early onset colorectal cancer (EOCRC)) is increasing throughout the world. This has predominantly been an increase in distal colonic and rectal cancers, which are biologically similar to late onset colorectal cancer (LOCRC) but with higher rates of mucinous or signet ring histology, or poorly differentiated cancers. The epidemiology of this change suggests that it is a cohort effect since 1960, and is most likely driven by an environmental cause. We explore the possible role of microplastics as a driver for this change. **Review:** The development of sporadic colorectal cancer is likely facilitated by the interaction of gut bacteria and the intestinal wall. Normally, a complex layer of luminal mucus provides colonocytes with a level of protection from the effects of these bacteria and their toxins. Plastics were first developed in the early 1900s. After 1945 they became more widely used, with a resultant dramatic increase in plastic pollution and their breakdown to microplastics. Microplastics (MPs) are consumed by humans from an early age and in increasingly large quantities. As MPs pass through the gastrointestinal tract they interact with the normal physiological mechanism of the body, particularly in the colon and rectum, where they may interact with the protective colonic mucus layer. We describe several possible mechanisms of how microplastics may disrupt this mucus layer, thus reducing its protective effect and increasing the likelihood of colorectal cancer. **Conclusions:** The epidemiology of increase in EOCRC suggests an environmental driver. This increase in EOCRC matches the time sequence in which we could expect to see an effect of rapid increase of MPs in the environment and, as such, we have explored possible mechanisms for this effect. We suggest that it is possible that the MPs damage the barrier integrity of the colonic mucus layer, thus reducing its protective effect. MPs in CRC pathogenesis warrants further investigation. **Future directions:** Further clarification needs to be sought regarding the interaction between MPs, gut microbiota and the mucus layer. This will need to be modelled in long-term animal studies to better understand how chronic consumption of environmentally-acquired MPs may contribute to an increased risk of colorectal carcinogenesis.

## 1. Introduction

Colorectal cancer (CRC), the third most common cancer worldwide (1.9 million cases in 2020), is increasing in those under 50 years of age [1,2]. This changing pattern of disease has been described in many countries [3], along with the observation of a predominant increase in left-sided disease (rectal, rectosigmoid and sigmoid cancers) [2]. The magnitude of this shift is demonstrated in a recent study from New Zealand, where the incidence of distal colonic cancer in men increased by 14 per cent per decade (incidence rate ratio (IRR 1·14), 95 per cent C.I. 1·00 to 1·30; *p* = 0·042); while the incidence of rectal cancer in men increased by 18 per cent (IRR 1·18, 1·06 to 1·32; *p* = 0·002) and that in women by 13 per cent (IRR 1·13, 1·02 to 1·26; *p* = 0·023). The magnitude of this change is consistent across a number of countries [3]. With this trajectory, it is estimated that within 10 years, 1 in 4 rectal cancers may be in those under 50 years of age [2].

The etiology of this epidemiological change is unknown [4]; however, the birth cohort appearance suggests that it is likely to be an environmental influence. Individuals born after 1960 have an increased risk of early-onset colorectal cancer compared to previous generations [5], suggesting that shared risk factors common across a generation are likely to be contributing. A wide variable of loosely associated factors has been suggested to explain this trend, such as changing trends in obesity, sedentary lifestyles, smoking, the Westernization of diets, and/or the use of antibiotics in early life [2]. The introduction of the contraceptive pill in the late 1950s [6] is another potential risk factor. However, no causal relationship or mechanistic link to any specific factor has yet been established [7].

The etiology of EOCRC, and sporadic colorectal cancer in general, is likely to be associated in some way with the gut microbiota and their interaction with the gut mucosa [8,9,10,11,12]. The mucosal epithelium is normally protected by a mucus layer that increases in thickness proximally to distally in the healthy colon [8], and provides a first line of defense to protect colonocytes from luminal bacteria [9]. The colonic mucus layer is made up of an inner sublayer, renewed from the secretions of the goblet cells at the luminal border, which expands distally into an outer mucus layer. While the inner mucus layer forms a size-exclusion barrier that separates colonic bacteria from the host epithelium [10], the outer mucus layer is recognized as providing a nutrient-rich habitat for some of these bacterial species [11]. The presence of certain bacteria in this layer has been shown to increase the risk of subsequently developing CRC [12].

Microplastics (MPs) that enter the diet and reach the colon have the potential to alter the existing balance between the gut microbiota and the mucus layer. As such, they may also alter the exposure of the colonocytes to various potentially harmful components of the gut microbiota, and therefore influence the incidence of CRC. In addition, given the propensity of MPs to adsorb hydrophobic molecules from the environment [13], the concept of direct exposure of colonocytes cells to MP-associated carcinogens may further increase risk (see below).

## 2. Microplastics

Plastic was first synthesized in the early 1900s and became more widely available after World War 2 [14]. Since then, plastic pollution, which was first recognized in the oceans in the early 1970s in studies on plankton, has reportedly increased from 2.3 million tons (2.1 × 10^6^ ton) in 1950 to 348 million tons (4.1 × 10^8^ ton) in 2017 [15], while the global release into the environment from primary MPs is estimated to be in the order of 3.2 million tons/year (2.9 × 10^6^ ton/year) [16]. This trajectory is unlikely to change in the foreseeable future (Figure 1).

**Microplastic composition.** In a purely chemical context, plastics per se are biochemically inert polymers. However, many plastics contain plasticizers (e.g., nonylphenols) which are important to confer specific properties (e.g., malleability)—these plasticizers are not chemically bonded to the plastic polymer, and can therefore leach into the environs of the plastic. This is exemplified by the inflammatory response that surgical mesh can induce when introduced into tissues [18]. The chemical structure determines the properties of plastics, such as durability, which renders them highly resistant to natural processes of complete degradation. It is this durability and stability of plastics that has led to their accumulation in the environment and, subsequently, their presence in the human food chain (Figure 2).

Primary MPs are those purposefully produced at less than 5 mm in length for commercial use (e.g., plastic pellets, synthetic textiles, microbeads), whereas secondary MPs are the result of environmental degradation of larger plastics such as plastic bottles, bags, and packaging [19]. Irrespective of whether MPs are primary or secondary, their uptake and subsequent bioaccumulation *in* the human body are increasingly considered to negatively impact the body’s usual mechanisms of damage repair, with resultant increases in apoptosis, necrosis, inflammation, oxidative stress, and aberrant immune responses [20].

These studies do not, however, consider the surface composition of ingested MPs, which may also alter their biological effect. Additives such as plasticizers, which can also bind to the surface of MPs, can be transported directly and/or alter the surface chemistry, allowing MPs to act as carriers for toxic chemicals. As such, the surface composition is likely to contribute to the health impacts of MPs; notably, the ability to accumulate in the colon and/or increase the risk of colorectal carcinogenesis [19].

**Exposure to microplastics.** The ingestion of MPs by humans starts at an early age [21,22] and continues throughout life [23,24,25]. While it is estimated to be 0.1–5 g/week in adults [26], in young children the average daily consumption of MPs from plastic infant feeding bottles was 1,580,000 particles, corresponding to an infant exposure 2600 times greater than adult consumption from water, food, and air combined [27]. For comparison, an unrelated study estimated annual adult MP consumption to be 48,000–52,000 MP per person, increasing to 94,000–114,000 MP with the inclusion of total inhaled particles [28].

The long-term effects of exposure to such high concentrations of MPs at a young age is yet to be determined. Likewise, despite the abundance of potential sources, little is known about the health impacts of MPs; notably, the potential implication(s) of MP bioaccumulation in the gastrointestinal (GI) tract over time. This is reinforced by a recent study that reports a finding of MPs in CRC patient colectomy samples [29]. These parallel developments raise the question of whether the increasing trend of young onset CRC with unknown etiology and the increasing pollution of plastic debris are linked.

A worldwide increase in BMI is increasingly seen as a risk factor for EOCRC [30]. In low- and middle-income countries, this is likely being driven by the consumption of diets dominated by higher intakes of processed fats and sugars compared to fresh foods [31], influenced by urbanization [32] and socio-economic status [33]. In contrast, it is the disadvantaged [34] and those more likely to be influenced by social inequality [33] that are more likely to develop diet-related obesity in high-income countries. Packaging may, in time, prove to be an underappreciated risk factor linking increased consumption of fast foods worldwide with an increased risk of EOCRC [19].

## 3. The Human Physiology of MP

The finding of large quantities of MPs in stool samples suggests most MPs transit directly through the small bowel into the colon. However, emerging evidence of MPs in human blood and tissue samples suggests that not all ingested MPs pass directly through the GI tract. Instead, given the observation that the distribution of particles in the respiratory tract relies heavily on particle size, it is likely that size may similarly influence the distribution, deposition and/or accumulation of MPs at this site. For example, while smaller particles may intuitively be considered to be of higher toxicity due to their high surface area/volume ratio, studies in zebrafish show that intestinal damage (evidenced by cracking of villi and splitting of enterocytes) is associated with exposure to larger (1 µm) rather than smaller (0.1 µm) MPs [35]. Oral exposure of mice to MPs, however, reportedly induces greater changes in gut microbiota in animals fed smaller (0.5 µm) rather than larger (50 µm) MPs [36].

Additionally, the porosity and/or surface-associated molecules are likely to influence the trafficking, distribution and/or uptake of MPs along the GI tract. Most plastics have hydrophobic surfaces because of the nature of their polymers (e.g., polystyrene). The hydrophobic nature of the surface of MPs allows adsorption (by hydrophobic interactions) of hydrophobic (e.g., the carcinogen benzo[a]pyrene) or partially hydrophobic (e.g., endocrine-disrupting phthalates) environmental pollutants or chemicals used in the manufacture of the plastic (e.g., styrene monomer is often bound to polystyrene). In short, because of the chemical nature of the surface of MPs, they can adsorb hydrophobic compounds (some of which are carcinogens), bind charged molecules and ions (including toxic metals) by electrostatic interactions, adhere microorganisms (some of which may be pathogens), or simply initiate a local inflammatory response (which may lead to non-genotoxic carcinogenesis). Environmental “weathering” may further contribute to the altered structural properties and surface porosity of MPs. All of these MP-mediated exposures have attendant health risks.

Mechanical forces within the gastrointestinal lumen and the effect of gastrointestinal fluids likely contribute to fragmentation of ingested MPs [37,38,39,40,41,42]. In addition, the chemical microenvironment (e.g., pH) along the colon is also likely to change toxic chemical adsorption characteristics, something that is likely to be particularly relevant with regards to the ingestion of weathered MPs. A few smaller MPs that are absorbed in the small intestine and end up in bile [43] also enter the colon as bile-associated MPs [44].

**MPs and the mucus layer.** In the process of passing through a body, MPs will come into contact with the loosely adherent outer mucous layer that provides a first line of defense in the healthy colon. Hydrophobic domains in this layer have the potential to bind MPs, some of which may subsequently be shed as a result of the physiological peristaltic functions of the gastrointestinal tract that results in the constant turnover of this layer [45]. Environmental and/or lifestyle factors that result in a thinner outer mucus layer [11] may instead facilitate the accumulation (and persistence) of MPs adjacent to the adherent inner mucus layer that covers the colonic epithelium [10].

The inner mucus layer, which normally functions as a barrier to protect the underlying epithelial cells from bacteria and toxic chemicals, is also likely to afford the intestinal epithelium a similar degree of protection from MPs. The accumulation of MPs adjacent to this inner layer may be aided by the development of a biofilm (see below). In this setting, close proximity of MPs with adsorbed carcinogens to colonocytes may lead to transfer of the carcinogens or pinocytotic uptake of MPs as well as their adsorbed carcinogens, resulting in the exposure of these cells to carcinogens.

**MPs and biofilm formation**. MPs that reach the inner colonic mucus layer may act as rafts for the development of biofilms, which are complex bacterial communities recognized as important modulators of gut health [46]. Evidence of bacteria-rich biofilms on the surface of MPs recovered from seawater [47,48] supports the idea that bacterial adherence to the surface of MPs in situ may provide a platform for nascent biofilm development. A similar scenario may also occur in the colon, facilitated by the attachment of protein and amino acids to the surface of negatively charged MPs [49]. Even small numbers of bacteria attached to MPs have the ability to start producing the polymeric matrix that forms the basis of a biofilm, affording the resident microbiota with a degree of protection from the external environment in the GI tract [50,51]. In turn, the development of biofilms has the potential to alter the physical and chemical properties of the MPs [47,52]. While biofilm communities are not necessarily associated with pathogenesis [51,53], the development of a MP-enriched biofilm that affects the integrity of the inner mucus layer may also be associated with the loss of epithelial barrier function.

In this setting, friction between MPs and the inner adherent mucus layer may elicit mechanical damage and/or trigger cell signaling pathways, promoting inflammation that has the potential to compromise barrier function [54]. Mice fed MPs have decreased *Muc2* gene expression [36,55]. Decreased *Muc2* expression is also associated with an increased risk of colorectal carcinogenesis in mice [56]. The possible link between ingestion of MPs and increased risk of carcinogenesis via expression of host mucin proteins gains significance with the observation that MUC2 is the predominant secreted gastrointestinal mucin in humans [57].

**MPs and loss of barrier function**. Paracellular movement of microparticles through enterocyte monolayers is associated with loss of transepithelial resistance and associated loss of tight junction function [58,59]. Loss of tight junction function is reported in the ileum of rats following ingestion of latex particles [60]. Transport of MPs into intraepithelial spaces via M cells that lack mucous and glycocalyx [61] may trigger this response, facilitated by physiological differences between the small and large intestine [62]. A high-fat diet (HFD) that triggers dysbiosis in mice also causes loss of intestinal barrier function [63], suggesting that high levels of dietary fat may have an additive effect with regards to loss of colonic epithelial barrier function when MPs are ingested. One consequence of MP-associated loss of epithelial barrier function is that it provides the means for colonic bacteria to breach this barrier.

## 4. MPs and Carcinogenesis

The idea that continuous exposure to MPs likely causes harm to humans has broadened to include consideration of how plastics can induce carcinogenesis in humans [64]. The carcinogenic potential of MPs has been the subject of two recent reviews [65,66], both of which conclude that an association likely exists. However, as Domench and colleagues point out, studies to date have mostly involved in vitro models and/or short-term studies in rodents, and therefore preclude clear conclusions [66].

Carcinogenesis can be either genotoxic or non-genotoxic [67]. Genotoxic carcinogenesis involves direct DNA damage caused by mutagens. MPs can carry environmentally acquired surface-associated toxic chemicals that include persistent organic pollutants (POPs), polycyclic aromatic hydrocarbons (PAHs), and hydrophobic organic chemicals (HOCs) [20,68,69]. This relates to the chemical nature of the surface of MPs, enabling them to adsorb hydrophobic compounds (some of which are carcinogens), bind by electrostatic interactions, charged molecules and ions (including toxic metals), and pathogenic bacteria. All of these MP-mediated exposures have attendant health risks that include the direct delivery of MP-associated toxic chemicals to the underlying epithelium [54]. The increased risk of biofilm-associated carcinogenesis [70] may reflect the close proximity a biofilm affords MPs to the intestinal epithelium.

MPs may also provide a vector for the delivery of carcinogenic bacterial toxins to the colonic epithelium. This is exemplified by colonic carriage of *Escherichia coli* that are associated with an increased risk of CRC [71] via expression of a genotoxin [72]. *E. coli* normally resides in the colonic lumen [73], but evidence of *E. coli* binding to MPs in an aquaculture model [74] suggests these bacteria may likewise bind to MPs in the colon. If so, MPs with adherent *pks+ E. coli* have the potential to act as a vehicle for the delivery of these genotoxic bacteria to the surface of the colonic epithelium, although there is evidence to suggest this is dependent on the loss of an intact inner mucus layer [75].

In simple terms, a dietary-related shift in the relative abundance of bacterial species (dysbiosis) that degrade colonic mucus [76] may increase the relative abundance of *pks+ E. coli* adjacent to the colonic epithelium. However, mice continuously exposed to PS MPs in drinking water for six weeks also showed evidence of colonic dysbiosis [36,55]. This gains significance with the increasing number of colonic microbiota now considered to have a role in the genesis of CRC [77,78]. While these bacterial species differ with regards to their respective pro-carcinogenic mechanisms [78], the idea that the association of these bacteria with MPs may facilitate the delivery of the respective bacterial toxins to the colonic epithelium is compelling, and would support the hypothesis that that long-term carriage of toxigenic bacteria may drive carcinogenesis in an otherwise healthy colon [79].

Non-genotoxic carcinogenesis can occur as a result of errors during cell division. In this setting, MPs may cause inflammation (see below) and/or deliver estrogenic compounds that interact with estrogen receptors on the cell surface, leading to increased cell proliferation and signaling non-genotoxic carcinogenesis. This gains significance with the understanding that some intestinal epithelial cells express estrogen receptors, that dietary estrogens and estrogen mimics may be colorectal cancer promoters [80], and that some plasticizers are estrogen mimics (e.g., 4-nonyl phenol) [81].

Further, the damaged mucus layer may enable MPs to translocate through the colonocytes to the lamina propria, where they are phagocytosed by intestinal macrophages that exist at this site [82]. MPs may simply act as a vector to deliver particle-associated toxins to the macrophages (see above). However, endocytosis of MPs by macrophages may also deliver surface-associated bacteria. Intestinal macrophages play an essential role in maintaining mucosal homeostasis and epithelial barrier integrity that is informed by macrophage polarization, a process by which the stimulus and/or environment dictates the functional end. Activated macrophages exhibit phenotypic plasticity, and a shift towards a predominantly M1 phenotype has the potential to perpetuate intestinal inflammation that is, in turn, associated with increased risk of carcinogenesis [83]. This hypothesis is informed by the observation that macrophages that phagocytose MPs undergo a metabolic shift to glycolysis as the means to provide cellular energy [84], illustrating how the phenotypic plasticity of macrophages may be influenced by chronic exposure to ingested MPs. It also supports the idea that chronic MP ingestion likely has an underappreciated role as a driver of inflammation-associated colorectal neoplasia.

MPs of varying sizes, concentrations, physical and chemical characteristics, and time of exposure can lead to the generation of reactive oxygen species (ROS) [85], with the resultant oxidative stress potentially providing a mechanism linking chronic MP-associated macrophage activation and resultant inflammation with resultant increased risk of CRC [64]. MP-mediated delivery of surface-associated toxic chemicals and/or bacteria to the macrophages may drive this process, but there is also evidence that MPs may not require surface-associated toxins to elicit an inflammatory response. Macrophages ingesting pristine MPs are shown to undergo a metabolic shift that is associated with altered cell surface markers and cytokine gene expression [84]. While increased expression of proinflammatory cytokines is shown to trigger the expression of mucins, aberrant glycosylation of these proteins may contribute to a defective mucus layer [86]. This may facilitate the production of reactive oxygen species (ROS) that are highly reactive with other molecules, including DNA and proteins [87]. Thus, MP ingestion that results in ROS-mediated oxidative damage to host DNA has the potential to promote oncogenic signaling and an increased risk of carcinogenesis.

## 5. Challenges and Knowledge Gaps

Many of the mechanisms described here are yet to be demonstrated in humans. For example, further clarification is needed regarding the interaction of the mucus layer, MPs and the gut microbiota. Additionally, most animal exposure studies have been completed over shorter study periods, thus longer-term studies are needed [66].

The global magnitude of plastic waste in the sea is difficult to measure; the conversion of this waste to MPs in the marine environment is known but unquantified, and human MP intake is extremely variable, including composition as well as particle numbers. Factors such as geographical location, dietary intake, and lifestyle can significantly influence MP exposure, the impact of which requires further study.

## 6. Conclusions

In this review, we explored the evidence for a possible link between MPs and an increase in CRC in those under 50. This changing epidemiology of CRC matches the time sequence that we might expect to see as an effect of rapid increase MPs in the environment and, as such, we have explored possible mechanisms for this effect. We described several possible mechanisms of how microplastics may disrupt the colonic mucus layer, thus reducing its protective effect, and thus increasing the likelihood of colorectal cancer.

Due to most aspects of human life being inoculated with plastics, the potential health effects of plastic exposure is huge; whether it be due to potential impacts on our health habits, or simply a confounder of our plastic use and consumption, MPs in CRC pathogenesis warrants further investigation. To achieve this, further research and inquest is required to assess and take steps towards reducing any potential adverse health effects and cancer risks in humans.

## Figures and Tables

**Figure 1 cancers-15-03323-f001:**
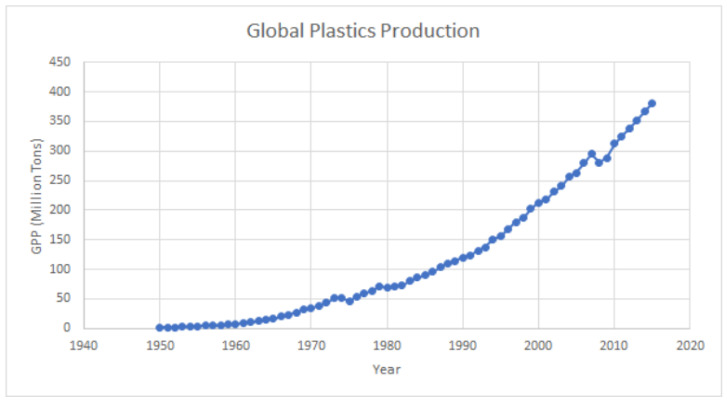
Global plastics production (1950–2015). Plastic production refers to the annual production of polymer resin and fibers. Sources: Ritchie, Hannah, and Max Roser. “Plastic pollution.” *Our World in Data* (2018) [17]. Copyright 2018, OurWorldInData.org, accessed on 12 June 2023.

**Figure 2 cancers-15-03323-f002:**
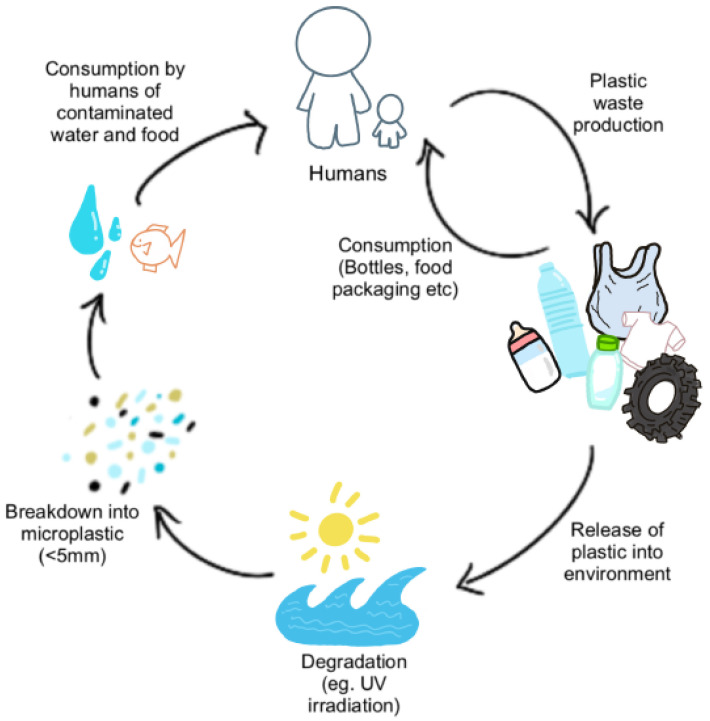
Cycle of Microplastics and Human Consumption. MPs are released into the environment as waste products from human consumption (plastic bags, bottles, textiles, tyres, etc.) where they undergo degradation and fragmentation from mechanical forces and UV irradiation. Some MPs are directly consumed by humans through leeching into food and water from food packaging and air pollution. MPs in the environment enter water sources and food supplies (e.g., seafood and livestock), thereby leading to consumption by humans.

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
