# Peer review of "Could Microplastics Be a Driver for Early Onset Colorectal Cancer?"

_cancers, 2023, doi:10.3390/cancers15133323_

Round 1

Reviewer 1 Report

This is a very interesting and timely review of the evidence for a possible link between MP and early onset colorectal cancer.  The authors provide a very nice overview of microplastics and their potential impact at a cellular level.  However, it seems that many of the conclusions are a bit of a leap from laboratory experiments, minimal animal studies and few correlational studies in humans.  However, this is often the case in the development of new hypotheses for association studies.  It would be useful for the authors to clearly state that many of the mechanisms being described have not yet been fully demonstrated in a human model.  

The manuscript was rather long and while i would not recommend removing particular sections, it may be helpful to add more subheadings.  Also are there possibly other covarying factors that are worth discussing.  At the start of the article several exposure shifts were mentioned - could the function of MP be due to cooccurring with these other environmental changes? 

Finally the manuscript lacked in human or population data that may add to the strength of the argument.  Is there any ability to correlate higher MP exposure (or potential for exposure) in areas of the world with higher EOCRC?  While rates may be changing worldwide, they are changing more rapidly or dramatically in particular areas - are these also areas more likely to have higher or longer exposure to MP?

The English is adequate but there is a need for close editing.  In many places throughout the manuscript there are missing words or duplicated words.  

Author Response

This is a very interesting and timely review of the evidence for a possible link between MP and early onset colorectal cancer.  The authors provide a very nice overview of microplastics and their potential impact at a cellular level.  However, it seems that many of the conclusions are a bit of a leap from laboratory experiments, minimal animal studies and few correlational studies in humans.  However, this is often the case in the development of new hypotheses for association studies.  It would be useful for the authors to clearly state that many of the mechanisms being described have not yet been fully demonstrated in a human model.  

Response: We thank the reviewer for this suggestion and have taken the liberty of adding their sentence at the beginning of the section titled “Challenges and Knowledge Gaps”. Please also see our response to Reviewer 2, acknowledging that studies to date as yet preclude clear conclusions.

The manuscript was rather long and while i would not recommend removing particular sections, it may be helpful to add more subheadings.  Also are there possibly other covarying factors that are worth discussing.  At the start of the article several exposure shifts were mentioned - could the function of MP be due to cooccurring with these other environmental changes? 

Response:  We agree with the reviewer that the manuscript is rather long. Our original submission was notably shorter. The word count was increased at the request of the journal.

Subheadings have been added throughout the manuscript.

The reviewer raises a good point with regards the possibility of covarying environmental changes. We have included the following, which we consider goes some way to address this.

A worldwide increase in BMI is increasingly seen as a risk factor for EOCRC [32]. In low- and middle-income countries this is likely being driven by the consumption of diets dominated by higher intakes of processed fats and sugars to fresh foods [33], influenced by urbanisation [34] and socio-economic status [35]. In contrast, it is the disadvantaged [36] and those more likely to be influenced by social inequality [35] that are more likely to develop diet-related obesity in high-income countries. Packaging may, in time, prove to be an underappreciated risk factor linking increased consumption of fast foods worldwide with increased risk of EOCRC [20]. 

  1. Finally the manuscript lacked in human or population data that may add to the strength of the argument.  Is there any ability to correlate higher MP exposure (or potential for exposure) in areas of the world with higher EOCRC?  While rates may be changing worldwide, they are changing more rapidly or dramatically in particular areas - are these also areas more likely to have higher or longer exposure to MP?

Response: Please see above.

  1. The English is adequate but there is a need for close editing.  In many places throughout the manuscript there are missing words or duplicated words.  

Response: We have reviewed the manuscript and addressed this comment.

Reviewer 2 Report

Looking into the increasing exposure to emerging environmental pollutants such as micro/nanoplastics in the environment and their potential health impact on humans, this perspective tries to enlighten readers on the possible mechanisms of early onset of colorectal cancer due to microplastics exposure. The manuscript is interesting and well written, however, it needs some minor revision. The details are given below:

Line 24: delete "layer" as it was written 2 times.

Line 26-27: Should be "...rapid increase of MPs..."

Line 59-60: Please bring clarity to the sentence.

Line 108: "in"..(non-italic).

Line 167:  Please bring clarity to the sentence.

The authors can also discuss the possible carcinogenicity due to the genotoxic effect induced by microplastics under the section "4. MPs and carcinogenesis". There are literatures in this direction.

Overall the Manuscript may be accepted after the incorporation of these minor revisions.

Minor English language revision is requested to make some sentences more clear. 

Author Response

Looking into the increasing exposure to emerging environmental pollutants such as micro/nanoplastics in the environment and their potential health impact on humans, this perspective tries to enlighten readers on the possible mechanisms of early onset of colorectal cancer due to microplastics exposure. The manuscript is interesting and well written, however, it needs some minor revision. The details are given below:

Line 24: delete "layer" as it was written 2 times.

Line 26-27: Should be "...rapid increase of MPs..."

Line 59-60: Please bring clarity to the sentence.

Line 108: "in"..(non-italic).

Line 167:  Please bring clarity to the sentence.

The authors can also discuss the possible carcinogenicity due to the genotoxic effect induced by microplastics under the section "4. MPs and carcinogenesis". There are literatures in this direction.

Response: We thank this reviewer for directing us towards this literature. We have added the following introduction to the section on MPS and carcinogenesis.

The idea that continuous exposure to MPs likely causes harm to humans has broadened to include consideration of how plastics can induce carcinogenesis in humans [66]. The carcinogenic potential of MPs has been the subject of two recent reviews [67, 68], both of which conclude that an association likely exists. However, as Domench and colleagues point out, studies to date have mostly involved in vitro models and/or short-term studies in rodents and therefore preclude clear conclusions [68].

Overall the Manuscript may be accepted after the incorporation of these minor revisions.

Minor English language revision is requested to make some sentences more clear. 

Response: we have reviewed the manuscript and revised language to make sentences clearer.